# Sexual Harassment Experiences, Knowledge, and Coping Behaviors of Nursing Students in Taiwan During Clinical Practicum

**DOI:** 10.3390/ijerph17134746

**Published:** 2020-07-01

**Authors:** Ting-Shan Chang, Ya-Ling Tzeng, Yu-Kuei Teng

**Affiliations:** 1Department of Public Health, China Medical University, Taichung 40402, Taiwan; u104077804@cmu.edu.tw; 2School of Nursing, College of Health Care, China Medical University, Taichung 40402, Taiwan; 3Department of Nursing, China Medical University Hospital, Taichung 40402, Taiwan

**Keywords:** sexual harassment, nursing students, gender, clinical practicum, experience, coping behaviors

## Abstract

Sexual harassment not only endangers nursing students’ physical and mental health but also considerably affects their future willingness to engage in the field of nursing. To identify experiences, knowledge, coping behaviors, and determinants of sexual harassment among nursing students during clinical practicum, this study conducted a cross-sectional survey where a structured self-report questionnaire was used. A total of 291 senior nursing students were recruited from four universities in Central Taiwan. Sixty-six nursing students (22.7%), including 59 women (23.3%) and 7 men (18.4%), reported experiencing sexual harassment during clinical practicum. Male students scored significantly higher than female students did on knowledge of sexual harassment (*p* = 0.028). Female students scored significantly higher than male students did on attitudes toward preventing and coping with sexual harassment (*p* = 0.05). Nursing students who were older, had fathers who had higher education levels, or had undergone gender-related courses were more likely to experience sexual harassment. More than one-fifth of nursing students experienced sexual harassment during their clinical practicum, making this a formidable challenge in nursing education. Education is required to prevent sexual harassment and enhance gender sensitivity among nursing students, who are at a greater risk of experiencing sexual harassment in clinical practicum.

## 1. Introduction

Sexual harassment is defined as unwelcome sexual behavior, including its verbal, physical, psychological, or visual forms, not only involving sexual behavior but also related to uneven power [1,2]. In this study, sexual harassment occurred in the medical context. For nursing students, sexual harassment during clinical practicum may lead to deterioration in physical and mental health as well as in a decline in willingness to take nursing jobs in the future [3,4]; this has become a crucial nursing education concern. When attending a patient, nursing staff are often in close or even physical contact with the patient; for example, checking vital signs, assisting in changing positions, changing wound dressings, etc., which increases the possibility of being harassed. Studies have shown that the proportion of patients or family members harassing registered nurses is 18–69.1% [5,6,7]. Therefore, nurses are considered to be at a high risk of workplace sexual harassment [5,8].

Incidences of the sexual harassment of nursing staff have been reported to be 37–72% [6,9] and some researchers have even revealed that approximately one quarter of nursing staff worldwide have experienced sexual harassment [10,11]; this is one reason hospitals have become a hostile working environment, resulting in the loss of considerable nursing staff. This has also become a cross-border and cross-cultural nursing professional matter that has extensive influences [12]. Previous studies have reported the prevalence rate of sexual harassment among nursing students to be 18–90% [1,7]. Nursing students with less social and clinical experience or with relatively low awareness are more likely to become victims of sexual harassment in the medical system. Nursing students are known to experience frustration, anger, helplessness, fear, and anxiety when sexually harassed [8], and these negative emotions inhibit their learning. However, compared with nursing staff, research on the sexual harassment experience of nursing students is limited, very few studies have focused on the sexual harassment experience of different gendered nursing students, and some of the research focusing on female nursing students were conducted long ago [13,14]. In more than fifteen years, no research has examined the sexual harassment encountered by Taiwanese nursing students during their internship, indicating that greater understanding of the situation is required if effective improvements are to be implemented on the basis of relevant factors.

The number of men in the nursing profession has increased yearly around the world. For instance, the proportion of US male nurses was reported to have increased from 6.6% in 2013 to 9.1% in 2017 [15]. In 2008, the proportion of male nurses in Japan was 4.8%, and by 2016, this proportion had risen to 6.9% [16]. In Taiwan, 6288 and 3316 male nursing students were enrolled in 2017 and 2010, consisting of 10.8% and 5.7% of all nursing students, respectively; the growth rate nearly doubled during this period [17]. The current situation and gender difference in sexual harassment during clinical practicum thus remains a matter for discussion. As mentioned earlier, although extant studies provide understanding of the sexual harassment that occurs during nursing clinical practicum, they are relatively old and lack gender analysis data [1,2,5,9]. Relevant surveys can serve as a source of understanding and enable an analysis of the influential factors to enable development of effective countermeasures. Additionally, such surveys and findings could attain the attention of teachers and personnel in the nursing field and are beneficial to reducing the occurrence of sexual harassment and also its negative influence on nursing students. Accordingly, the purpose of the present study was to identify the sexual harassment experience, knowledge, and coping behaviors of nursing students and the influencing factors of such harassment during clinical practicum.

## 2. Materials and Methods

### 2.1. Study Design

A cross-sectional survey was conducted using a self-report structured questionnaire.

### 2.2. Setting and Participants

From June to December 2019, questionnaires were distributed at four universities with a nursing department in Central Taiwan. Nursing students with clinical practicum experience were recruited to be research participants. The recruitment criteria were as follows: (1) in the senior year, (2) clinical practicum experience within the current year of study. Those students who had not completed the majority of their clinical practicum, such as fundamental nursing, medical-surgical nursing, maternity nursing, pediatric nursing, and psychiatric nursing, were excluded.

### 2.3. Ethical Considerations

The study was approved by the Institutional Review Committee of the Affiliated Hospital of China Medical University in Taiwan (CMUH107-REC3-016) and a consent form was provided to each participant. The researcher ensured that all necessary information was provided, such as the purpose of the study, the procedures involved, anonymity and confidentiality, voluntary participation, and the freedom to withdraw from the study at any time, so that students could make informed decisions before signing the form. Only students with written informed consent could participate in the study. Participants were requested not to write down names or provided any form of identification on the questionnaire and the consent form to ensure confidentiality and anonymity. The completed questionnaires were locked and keyed, and the information provided by the participants were for the purpose of this study only. Meanwhile, collecting the experience of sexual harassment may make certain participants feel uncomfortable, so we also provided the contact information of the researcher to provide any necessary assistance, such as psychological counseling.

### 2.4. Data Collection

One teacher in each university was invited to assist in participant recruitment and questionnaire distribution. All of the four teachers were female and were not involved in research or internship coaching. The teachers had an hour of training before data collection and were provided with written guidelines to obtain consistency. They openly recruited students who met the inclusion criteria in the school and provided them with all necessary information such as the purpose of the study, the procedures involved, and the conditions of anonymity, confidentiality, and voluntary participation. A hardcopy questionnaire was distributed after the participant consent form was signed. It takes about 20 min to finish the questionnaire. After returning the questionnaire, students received a small gift of about $0.50. The present study distributed 367 questionnaires and retrieved 310 questionnaires, a response rate of 84.4%. Excluding 19 incomplete questionnaires, a convenience sample of 291 was collected.

### 2.5. Measures

We used the structured questionnaires for sexual harassment in the medical field developed by Kao and Wang as the measurement. The questionnaire was developed through rigorous steps and with reliability and validity [14].

#### 2.5.1. Demographic Characteristics

The demographic questionnaire concerned age, gender, parents’ level of education, and whether participants had taken gender-related courses, such as Gender, Culture and Society, and Gender and Health Care.

#### 2.5.2. Sexual Harassment Knowledge

A sexual harassment definition scale was employed to determine the nursing students’ knowledge of sexual harassment. The scale mainly tested participants’ understanding of the concept of sexual harassment, which consists of the definition of verbal and non-verbal harassment. There were a total of 23 questions: 8 verbal definitions, for example, deliberately connecting professional terminology to sex-related jokes, and 15 nonverbal definition questions, for example, using nursing activity to let the nurse do unnecessary physical observation or contact. A 5-point Likert scale indicating the level of agreement was employed for scoring: 4 points indicated strongly agree; 3, mostly agree; 2, partly agree; 1, slightly agree; and 0, completely disagree. The total score ranged from 0 to 92. A higher score suggested higher level of agreements regarding the definition of sexual harassment. The construct validity of the scale is 0.691–0.967, and the reliability (Cronbach’s α) is 0.96–0.99 [13].

#### 2.5.3. Sexual Harassment Experience

The subscale of sexual harassment behaviors and experience in medical situations was employed. Both verbal and nonverbal sexual harassment questions were present in the subscale, and the identity of the harasser, the department in which the harassment occurred, and the location of the harassment were asked in each question. Taking verbal harassment as an example, deliberately connecting professional terminology to sex-related jokes. When such sexual harassment occurs, the identity of the harasser is a male physician, female physician, male patient, female patient, male family member, female family member, or other medical staff (please fill in their identity). In which department did sexual harassment happen? Obstetrics and gynecology, urology, family medicine, rehabilitation, dermatology, psychiatry, surgery room, intensive care unit, burn center, ophthalmology, internal medicine, surgery, orthopedics, emergency department, and others (please fill in the department). Where did this sexual harassment occur? Ward, nursing station, clinic, home visitation, lounge, meeting room, operating room, or other (please fill in). A 4-point Likert scale was used for scoring, from 3 (occurs frequently) to 0 (never occurs), with the total score ranging from 0 to 69. A higher score suggested more experience in such types of sexual harassment. The construct validity of the scale is 0.330–0.835, and the reliability (Cronbach’s α) is 0.88–0.96 [13].

#### 2.5.4. Coping Behaviors against Sexual Harassment

The sexual harassment coping behaviors subscale [14] was employed. The participants indicated their actual or possible (if they had no experience of sexual harassment) coping behaviors when faced with sexual harassment. The subscale consists of 37 questions, each of which is posed in relation to verbal and nonverbal sexual harassment, leading to 74 questions in total. The coping behaviors were divided into four aspects, namely solving the problem, for example, I would move my body right away; redefining cognition, for example, I would report to the teacher, head nurse, registered nurse, physician or guard; adjusting emotion, such as, I would ignore it; and releasing emotion such as, I would mock the harasser in an ironic way. Each question was scored from 0 (never) to 4 (always), with the total score ranging from 0 to 296. A higher score implied greater likelihood of applying coping behaviors. The construct validity of the scale is 0.366–0.695, and the reliability (Cronbach’s α) is 0.93–0.94 [13].

#### 2.5.5. Attitude toward Sexual Harassment

The sexual harassment coping and preventative strategy subscale was employed. This scale is composed of 31 questions concerning improving an individual’s coping ability, for example, understanding the relevant laws and being able to take legal action; improving an individual’s cognition and monitoring system, for example, strengthening the surveillance system in medical institutions; and enhancing social organizations and policies, for example, establishing anti-sexual harassment campaign groups. Each question was scored from 0 (completely disagree) to 4 (strongly agree), with a total score ranging from 0 to 148. A higher score suggested greater agreement on the improvement of the aspect. The construct validity of the scale is 0.560–0.83, and the reliability (Cronbach’s α) is 0.88–0.98 [13].

### 2.6. Data Analysis

All statistical analyses were performed using SPSS (version 25.0, SPSS Inc. Chicago, IL, USA) We excluded invalid questionnaires (i.e., missing responses ≥20% or apparently unreliable responses—like filling the same response number for all items, or selecting more than one answer for one item). Missing values were replaced by the series mean for continuous variables. We initially calculated descriptive statistics to identify the characteristics of our participants. Chi-square tests were used to assess the relationship between gender and the knowledge, attitude, and behavior to exposure to sexual harassment. Then, we conducted logistic regression analyses with odds ratio between the influential factors and exposure to sexual harassment. All reported *p*-values are two-tailed and statistical significance was set at 0.05.

## 3. Results

### 3.1. General Characteristics of Participants

The research participants were mainly aged 20–24 years (96.2%); 38 were men (13.1%) and 253 were women (86.9%). Among them, 66 of the participants (22.7%)—7 men (18.4%) and 59 women (23.3%)—reported that they had experienced sexual harassment. Most of the participants had taken relevant gender-related courses (225; 77.3%). In addition, 264 (90.7%) and 27 (9.3%) of the participants had fathers who had an education level of below university and graduate school, respectively; the percentages for the participants’ mothers were 284 (97.6%) and 27 (2.4%), respectively (Table 1).

### 3.2. Experience of Sexual Harassment during Clinical Practicum

Overall, 66 of the 291 participants (22.7%) had experienced sexual harassment during the recent year of internship. The data revealed prevalence of 60.6% for verbal harassment, 13.6% for nonverbal harassment, and 25.8% for both verbal and nonverbal harassment. Among those who had experienced sexual harassment, 5 men (71.4%) and 35 women (59.3%) had experienced verbal sexual harassment; 9 women (15.3%) but no men had experienced nonverbal sexual harassment; and finally, 2 men (28.6%) and 5 women (25.4%) had experienced both verbal and nonverbal sexual harassment. Furthermore, the male and female nursing students had been harassed 21 (8.8%) and 219 (91.2%) times, respectively. The most and second most common harassers, for the male and female nursing students, were male patients (14 times, 66.7%; 125 times, 57.3%) and male doctors (3 times, 14.3%; 44 times 20.1%), respectively. For the male and female nursing students, the most and second most common hospital departments in which sexual harassment occurred were the departments of internal medicine (11 times, 52.4%; 60 times, 27.4%) and psychiatry (3 times, 14.3%; 45 times 20.5%), respectively. The most and second most common locations of sexual harassment for the male and female nursing students were the ward (13 times, 68.4%; 139 times, 70.6%) and nursing station (5 times, 26.3%; 24 times, 12.2%; Table 2).

### 3.3. Nursing Students’ Sexual Harassment Knowledge, Coping Behaviors, and Attitude

Table 3 reveals that the male and female nursing students scored 75.97 ± 25.13 and 65.57 ± 25.13, respectively on sexual harassment knowledge, indicating a significant gender difference (*p* = 0.028). The women obtained higher scores than the men in terms of all types of coping behavior (solving the problem, redefining cognition, adjusting emotion, or releasing emotion); however, the differences were non-significant. The female nursing students received a significantly higher score in attitude (99.07 ± 24.66) than the male nursing students (89.39 ± 8.04; *p* = 0.05). In the subscale, the women scored significantly more highly than the men only in questions concerning the enhancement of social organizations and policies (*p* = 0.01). Although the female nursing students obtained higher scores in improving competence and cognition, the difference was non-significant.

### 3.4. Relevant Factors Affecting Nursing Students’ Experience of Sexual Harassment in Medical Situations

Logistic regression was employed to determine the factors affecting nursing students’ experience of sexual harassment in medical situations, and the results indicated that awareness of sexual harassment increased with age. Those who had taken a gender-related course were 1.86 times more conscious of the possibility of sexual harassment. Those students whose father had a graduate school education were 2.97 times more conscious of the experience of sexual harassment (Table 4).

## 4. Discussion

The present study is the first survey research examining the sexual harassment experience, knowledge, coping behaviors, and attitude of different gender nursing students in Taiwan. The results revealed that 22.7% of the nursing student participants had experienced sexual harassment during their clinical practicum, which is similar to the results of a Korean study, which reported that 17.9% of nursing students have experienced sexual harassment [1], but our value is lower than the result found from Israeli research (90%) [7]. The sexual harassment rate obtained in the present study is also lower than that of a survey study conducted in 2001 that recruited only female nursing students (42.5–59.6%) [13,18]. This difference may have been due to social or cultural factors and affected by the Gender Equality Education Act and Sexual Harassment Prevention Act. In 2004 and 2005, Taiwan implemented the Gender Equality Education Act and Sexual Harassment Prevention Act, which stipulates that employers shall provide necessary prevention measures for employees. If there are more than 30 employees, it is necessary to establish measures of prevention, complaint, and punishment relating to sexual harassment. Anyone who violates the law must be brought to account. In order to promote gender equality, many schools also offer related gender studies courses, such as Gender, Culture and Society, Gender and Health Care. Therefore, we speculate that education and laws may be one of the reasons for reducing sexual harassment. However, this speculation still needs more research to confirm, and it is recommended that future research further explores the factors that reduce sexual harassment in the workplace as a basis for improvement.

Previous research has shown that 37–72% of nursing staff have suffered sexual harassment in the medical field. Compared with other professions, for example, female firefighters and female workers in the industrial sector report that they have suffered sexual harassment in the workplace at 21.7% and 12% respectively [19,20], the prevalence of workplace sexual harassment among nursing staff is obviously higher. This may be because nursing involves working physically and emotionally close to patients, doctors, medical attendants, informal caregivers, paramedical staff, and administrative members [9]. Although the prevalence of sexual harassment among nursing students is lower than that of nursing staff, it may be related to their low sensitivity to sexual harassment. In addition, in Taiwan, nursing students usually have teachers to guide them during clinical practicum, which may reduce the occurrence of sexual harassment. Nonetheless, more than one fifth of the nursing student participants in the current study had had sexual harassment experience during clinical practicum, which indicates that this is a nursing education matter that requires attention. Furthermore, preceding research has mostly focused on female nursing students, whereas the present study also included male nursing students. The results revealed that although sexual harassment of female nursing students was more common than that of male nursing students, a considerable percentage of both male and female nursing students have been harassed during clinical practicum. Workplace sexual harassment is not merely a sexual concern but indicative of an unequal power relationship [21]. Young, unmarried, or new, low-ranking nursing staff and nursing students easily become the subjects of sexual harassment [1,22,23]. The sexual harassment rate is generally higher among women than men. A survey conducted by the Ministry of Labor in Taiwan indicated that only 0.8% of their male sample claimed to have experienced workplace sexual harassment compared with 3.5% of the female sample, a factor of fourfold difference [24]. Nursing care is traditionally a profession for women; nursing is not a mainstream, masculine job for men. Accordingly, male nurses are more likely to have experienced sexual harassment than men in other professions [25]. The results of the present study have also reflected this phenomenon. Nursing educators should thus focus on bullying and sexual harassment problems that can occur during internships for both genders.

Further examination of gender differences revealed that 71.4% of male nursing students had experienced verbal sexual harassment, higher than the rate among female nursing students, who were more likely to experience nonverbal sexual harassment. This suggests that physical contact is more commonly involved in sexual harassment of female nursing students compared with male nursing students; hence, nursing teachers are advised to instruct their students to protect themselves before they begin their clinical practicum. Moreover, the majority of harassers of nursing students are male patients, and harassment mostly occurs on wards; the most and second most common departments in which harassment was discovered to occur were the departments of internal medicine and psychiatry, which is a similar finding to other relevant studies [13,18]. This may be related to nursing staff’s frequent contact with patients in these locations. These findings, including locations and units where sexual harassment is most common, can be used as a reference for future education of nursing students on sexual harassment prevention or development of improvement strategies. The difference between the scores in each scale demonstrated that the male nursing students obtained higher scores than the female nursing students in sexual harassment knowledge despite female nursing students being more likely to experience sexual harassment [7]. Accordingly, strengthening the knowledge and preventative ability of female nursing students is essential. The sexual harassment knowledge scores obtained by the men and women were only 82% and 71%, indicating that further improvements are required. In terms of attitude toward sexual harassment, the female nursing students received higher scores than the male nursing students, particularly in the aspect of enhancing social organization and policies. The possible reason for this is that women are conscious that they are more likely to experience harassment and therefore think that external forces should be strengthened. Follow-up studies should be conducted to clarify the reasons behind this different attitude between the genders.

The present study also discovered that older nursing students had a superior awareness of sexual harassment experience. The participants of the present study were young students with an average age of 22 years; thus, the age range in this study was not considerably large. However, a few of the students were older and had working experience, which may have given them a greater awareness when encountering sexual harassment. Furthermore, the nursing students who had taken a gender-related course were more aware of sexual harassment experience than those that had not taken such a course; this may have been because those who had taken such courses were more informed of information concerning sexual harassment. Some researchers reported that nursing education accentuates nurses’ caring role, and this may prevent students from decisively and firmly rejecting sexual harassment [8]. Kim et al. (2018) interviewed 13 nursing students who had experienced sexual harassment during internship and compiled 12 themes. Among them, being “unprepared to respond”, “lack of education,” and being “unsure about when behavior crosses the line” clearly showed that the students’ requirements for proper education and preparation for sexual harassment. We suggested that nursing educators should enhance sexual harassment prevention courses and develop materials applicable to this purpose to improve nursing students’ preventative and coping abilities, providing them with confidence when experiencing sexual harassment during internship and maintaining their learning and patient care quality.

## 5. Limitations

The present study only investigated universities in central Taiwan; hence, the results cannot be applied to other regions in Taiwan. It is appropriate to include more regions to obtain a larger sample and a more inferential result. In addition, the present results cannot be generalized to nursing students in other academic systems (e.g., junior colleges), because the research participants were recruited from universities only. However, the current education system in Taiwan is mainly universities, junior colleges are in the minority—so the participants of the present study are considerably representative. Besides, a total of 26,206 nursing students in Taiwan, of whom 3807 (14.5%) were male and 22,399 were female (85.5%) [17]. In this study, the rate of male and female nursing students were 13.1% and 86.9%, respectively. In general, this study cohort has a considerable degree of equivalence with the nursing student population in Taiwan, including age, gender, and required clinical practicum credits. Furthermore, a cross-section questionnaire was employed, which has limited ability to reveal the changes in students’ internship experience. Follow-up studies should be conducted for longitudinal research to gain more in-depth understanding of the sexual harassment experience of nursing students during clinical practicum and factors influencing sexual harassment experience.

## 6. Conclusions

The present study provides crucial information regarding the sexual harassment experience of nursing students during their practicum, particularly regarding male nursing students, who have not been emphasized in previous studies. The results revealed that sexual harassment was commonly experienced by nursing students of both genders. Gender-related education can help students identify occurrences of sexual harassment in the clinical environment; consequently, implementing preventative sexual harassment education before clinical practicum and enhancing students’ sensitivity to gender-related matters may improve students’ preventative and coping abilities, reduce the negative influence of sexual harassment, and ensure that high learning and patient care quality are maintained.

## Figures and Tables

**Table 1 ijerph-17-04746-t001:** Demographic characteristics and selected variables (*n* = 291).

Variable		Sexual Harassment *n* (%)		
		No	Yes	Total
Age	20–24	221 (98.2)	59 (89.4)	280 (96.2)
	25–30	4 (1.8)	3 (4.5)	7 (2.4)
	31–35	0 (0.0)	2 (3.0)	2 (0.7)
	36–40	0 (0.0)	2 (3.0)	2 (0.7)
Gender	Male	31 (13.8)	7 (10.6)	38 (13.1)
	Female	194 (86.2)	59 (89.4)	253 (86.9)
Father’s level of education	Uneducated or elementary school	10 (4.4)	1 (1.5)	11 (3.8)
	Junior high school	35 (15.6)	2 (3.0)	37 (12.7)
	Senior or vocational high school	86 (38.2)	23 (34.8)	109 (37.5)
	Junior college	38 (16.9)	12 (18.2)	50 (17.2)
	University	34 (15.1)	23 (34.8)	57 (17.2)
	Graduate school	22 (9.8)	5 (7.6)	27 (9.3)
Mother’s level of education	Uneducated or elementary school	10 (4.4)	2 (3)	12 (4.1)
	Junior high school	24 (10.7)	3 (3)	27 (9.3)
	Senior or vocational high school	106 (47.1)	25 (37.9)	131 (45)
	Junior college	41 (18.2)	16 (24.2)	57 (19.6)
	University	40 (17.8)	17 (25.8)	57 (19.6)
	Graduate school	4 (1.8)	3 (4.5)	7 (19.6)
Experience of gender education	No	45 (20)	21 (31.8)	66 (22.7)
	Yes	180 (80)	45 (68.2)	225 (77.3)

**Table 2 ijerph-17-04746-t002:** Experience of sexual harassment during clinical practicum (*n* = 66).

Variables	Times	Percentage	Variable	Times	Percentage
Male Nurses			Female Nurses		
Harasser	21	8.8	Harasser	219	91.2
Male doctor	3	14.3	Male doctor	44	20.1
Female doctor	0	0.0	Female doctor	7	3.2
Male patient	14	66.7	Male patient	125	57.1
Female patient	2	9.5	Female patient	9	4.1
Patient’s male family member	2	9.5	Patient’s male family member	24	11.0
Patient’s female family member	0	0.0	Patient’s female family member	7	3.2
Caregiver	0	0.0	Caregiver	2	0.9
Nurse	0	0.0	Nurse	1	0.5
Department in which sexual harassment occurred	21	8.8	Departments in which sexual harassment occurred	219	91.3
Gynecology	1	4.8	Gynecology	14	6.4
Urology	0	0.0	Urology	16	7.3
Family medicine	0	0.0	Family medicine	9	4.1
Physical medicine and rehabilitation	0	0.0	Physical medicine and rehabilitation	1	0.5
Dermatology	0	0.0	Dermatology	2	0.9
Psychiatry	3	14.3	Psychiatry	45	20.5
Operating room	0	0.0	Operating room	5	2.3
Intensive care unit	0	0.0	Intensive care unit	3	1.4
Burn center	0	0.0	Burn center	1	0.5
Ophthalmology	1	4.8	Ophthalmology	1	0.5
Internal medicine	11	52.4	Internal medicine	60	27.4
Location of sexual harassment	19	8.8	Location of sexual harassment	197	91.2
Ward	13	68.4	Ward	139	70.6
Nursing station	5	26.3	Nursing station	24	12.2
Outpatient clinic	0	0.0	Outpatient clinic	18	9.1
Conference room	0	0.0	Conference room	0	0.0
Lounge	0	0.0	Lounge	0	0.0
Operating room	0	0.0	Operating room	1	0.5

**Table 3 ijerph-17-04746-t003:** Participants’ scores for sexual harassment knowledge, coping behaviors, and attitude.

Variable		Male (*n* = 38)		Female (*n* = 253)		*p*
		Mean	SD	Mean	SD	
Definition of sexual harassment		75.97	25.13	65.57	34.24	0.028
	Verbal definition	24.92	8.78	21.50	11.25	0.036
	Nonverbal definition	51.05	17.52	44.07	23.63	0.033
Corresponding behavior		130.08	66.01	141.55	69.45	0.340
	Solving the problem	68.87	36.22	78.20	38.85	0.165
	Redefining cognition	36.11	20.35	37.37	21.81	0.738
	Adjusting emotion	19.26	14.77	19.94	16.26	0.810
	Releasing emotion	5.84	6.44	6.05	6.38	0.854
Attitude		89.39	28.04	99.07	24.66	0.050
	Improving competence	9.84	4.32	10.91	3.81	0.116
	Improving cognition	38.24	11.93	41.51	11.40	0.102
	Enhancing social organizations	41.32	13.65	46.66	11.55	0.010

**Table 4 ijerph-17-04746-t004:** Logistic regression analysis of the relevant factors affecting participants’ sexual harassment experience (*n* = 291).

Variable		OR	95% CI	p
Age		1.244	1.066-1.450	0.005
Sex	Male			
	Female	0.742	0.311-1.773	0.503
Father’s level of education	Uneducated or elementary school			0.008
	Junior high school	0.440	0.045–4.274	0.479
	Senior or vocational high school	0.251	0.045–1.410	0.117
	Junior college	1.177	0.402–3.446	0.767
	University	1.389	0.432–4.468	0.581
	Graduate school	2.976	0.985–8.994	0.053
Mother’s level of education	Uneducated or elementary school			0.209
	Junior high school	0.267	0.032–2.249	0.224
	Senior or vocational high school	0.167	0.024–1.135	0.067
	Junior college	0.314	0.066–1.495	0.146
	University	0.520	0.105–2.589	0.425
	Graduate school	0.567	0.114–2.809	0.487
Experience of gender-related education	No			
	Yes	1.867	1.012–3.444	0.046

Note. OR = odds ratio; CI = confidence interval.

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
