# Peer review of "Sexual Harassment Experiences, Knowledge, and Coping Behaviors of Nursing Students in Taiwan During Clinical Practicum"

_ijerph, 2020, doi:10.3390/ijerph17134746_

Round 1

Reviewer 1 Report

Overall evaluation and comments

This research paper covers an interesting topic in health professions education – exploring sexual harassment in nursing students in Taiwan. The authors investigate this topic by conducting a cross-sectional survey using a self-reported questionnaire.

While there are many interesting aspects to this study, there needs to be a significant amount of work done in the Introduction, Methods, Results sections to strengthen the robustness of the manuscript.

Title and Abstract

The title reflects the manuscript well. The abstract is well written overall.

Introduction

I found the introduction to this paper somewhat underdone. In a research paper about sexual harassment using a self-report questionnaire, I would expect the authors to define sexual harassment in the introduction as this concept is central to the research, yet this is not done. How is sexual harassment defined by the authors? How is it defined by the questionnaire used? Also there are a number of claims citing no evidence to support them (see below).

pg 1 lines 26-28  The authors state:

“When attending a patient, nursing staff are often in close or even physical 26 contact with the patient; therefore, nurses are considered to be at a high risk of workplace sexual harassment 27 [3,4]” – Does this imply that sexual harassment is coming predominantly from patients or other co-workers, or nursing students? This needs more clarification.

First introductory paragraph - the introduction mixes up both nursing student and nursing staff prevalence rates of sexual harassment, organising this data into reporting on both groups but separately would be useful to assist in understanding the background research literature, as well as to help with a logical argument for the research the authors are trying to make.

 pg 1 line 36 – The authors state:

 “However, research on the sexual harassment experience of nursing students is limited, and some of it was conducted long ago.” This statement needs to be supported with evidence. Even though the research on the sexual harassment experience of nursing students is limited, there is more research completed on this topic than the 12 references listed in the Reference section. Expanding on what has been done (even if it was a while ago) and where this research is filling the gap, would contribute to this study being more robust.

pg 1 lines 37-38 – The authors outline the need for this particular research project clearly, well done.

“In the past two decades, no research has examined the sexual harassment encountered by Taiwanese nursing students during their internship, indicating that greater understanding of the situation is required if effective improvements are to be implemented on the basis of relevant factors.”

pg 1 lines 40-41 The authors state:

“For instance, the 40 proportion of male nurses was reported to have increased from 2.7% in 1970 to 11% in 2017 [12]” I am assuming the authors mean world-wide? This needs some clarification.

pg 1 lines 43-45 – “However, research on sexual harassment has mostly considered female nurses or nursing students; few studies have investigated the gender differences in sexual harassment experienced by nursing students during internship.” These claims need citations to support them.

pg 1 lines 48-89 – “Although extant studies provide understanding of the sexual harassment that occurs during nursing clinical practicum, they are relatively old and lack gender analysis data.” These claims need citations to support them.

Methodological Rigour 

It is difficult to assess this article appropriately without copies of the questionnaire used (so I have written my review in this context). Thus, the questionnaire needs to be provided as an appendix. Was the questionnaire designed for nursing students, staff or both?

Definitions and explanations of all the subscales need to be given otherwise it is very difficult to interpret the results.

Was this an on-line or a hardcopy questionnaire? Am assuming hardcopy but this needs to be mentioned specifically. How long did the questionnaire take to complete?

What sampling technique was used?

pg 2, lines 62-3 - The recruitment criteria was: “(1) in the senior year, (2) clinical practicum 62 experience within the current year of study, and (3) willingness to participate in the questionnaire.” I am not familiar with “willingness to participate in the questionnaire” as a criteria for recruitment of participants. As depending on what this may mean, it would introduce a tremendous amount of bias into the study.

Ethical approval was undertaken which is excellent to see.

Explaining why certain demographics were used would be helpful e.g. why look at parents’ level of education? and What is a ‘gender-related course?’

Results

On the whole, the results were well represented, however due to the lack of definitions about the subscales it makes the results harder to interpret.

How well did the cohort that took the survey match the total cohort – i.e. was it representative? This important information is absent.

The ‘Experience of sexual harassment during clinical practicum’ section was interesting, I thought the data about where (by department) and with specific environments (e.g. the ward) was interesting to add, as it gives detailed information if one was to run an intervention.

Discussion and Conclusion

The Discussion section was well written overall, yet due to the introduction only covering a certain amount of the literature, how this study contributes to the wider literature as a whole could have been expanded on.

pg 4 lines 162-164 The authors state:

“This difference may have been due to social or cultural factors and affected by the Gender Equality Education Act and Sexual Harassment Prevention Act implemented in 2004 and 2005, respectively, in Taiwan.” This statement needs more explanation. Additionally the difference could also be because the studies could be using different questionnaires to test for the same thing, therefore the results are not comparable.

pg 4 lines 173-174 The authors state:

“Young, unmarried, or new, low-ranking nursing staff and nursing students easily become the subjects of sexual harassment.” These claims need citations to support them.

References

There are some typos as well as incomplete information given in the References section that need to be fixed (e.g. look at reference numbers 13 and 17).

Author Response

“請參閱附件。”

Reviewer 2 Report

This paper reports on sexual harassment experiences for a sample of 291 nursing students. The advantages of the present study is that it provides data for students in a profession in which they have a high risk of being sexually harassed, and it includes data for male students as well as female students and compares sexual harassment experiences by gender.

For an overall assessment, in my view, this paper provides interesting and useful information on how common it is to be sexually harassed at the start of a career in an occupation important for public health. It is not an exciting or innovative paper but the findings should raise awareness of the prevalence of a problem that is recognized to lead to turnover and negative health consequences for the victims.

I probably would have found this to be too narrow for a general journal and that is would be a better fit in a nursing or practitioner journal. Based only on the title of your journal, the topic of nursing seems to fit, but I am not familiar enough with your journal’s mission.

What I liked about the paper is that it is well written and clear about its points and results, and short, although it could be shorter still because there was some redundancy at the end.

It is relevant because sexual harassment has been an important concern for decades but has become even more important recently. The paper itself reads well and is interesting.

I have not tried to identify whether there are studies of nursing students generally or of nursing students in Taiwan. It has the advantage of including men in the survey. There is less empirical evidence on sexual harassment of men. The sample size for men is small so there is really little statistical power to identify significant differences even if they exist.

The conclusions are consistent with the evidence and arguments presented.

Suggestions:

The text of the paper refers throughout to Tables 1 – 4. But these four tables are not included in the main paper but in a separate file denoted as “non-published.” I am not sure what the journal policy is, but these tables should either be published with the text or else the text should indicate that the tables referred to are available somewhere else.

If space is a concern, Table 2 could be shortened or eliminated. It is thoroughly discussed in the text.

Data collection – it would be helpful to have a little more information about the data collection procedure. For example, were all teachers in a nursing department invited to assist in recruitment and questionnaire distribution? What share of teachers agreed to assist? If known, did the gender of the teacher have any relation to their participation or response rates?

Reviewer 3 Report

This is a sound research report that brings together an extensive survey that finds a shift in the Taiwanese nurses experience with sexual harassment. The authors are correct the new Taiwanese law on Harassment combine with companies training that this form of behavior is now unacceptable has  impact and is changing conventional sexual interaction. I was struck that a high number of incidence involve patient/nurse interaction. It would be revealing to know some of the verbal expressions or attempts, if any at touching. The problem with survey is their strength large number of respondents is their weakness the absence of rich ethnographic context. Still I am in complete agreement with the authors that their "study provides crucial information regarding the sexual harassment experience of nursing 314 students during their practicum, particularly regarding male nursing students, who have not been emphasized in previous studies." I would think other research woking on sexuality in China,  in the nursing profession or on the topic of sexual harassment would be interested in the study.